# Comparative Evaluation of the Antiglycation and Anti-α-Glucosidase Activities of Baicalein, Baicalin (Baicalein 7-*O*-Glucuronide) and the Antidiabetic Drug Metformin

**DOI:** 10.3390/pharmaceutics14102141

**Published:** 2022-10-09

**Authors:** Guglielmina Froldi, Francine Medjiofack Djeujo, Nadia Bulf, Emma Caparelli, Eugenio Ragazzi

**Affiliations:** Department of Pharmaceutical and Pharmacological Sciences, University of Padova, 35131 Padova, Italy

**Keywords:** BSA, AGEs, cellular uptake, ROS, flavonoids, HT–29 cells, diabetes mellitus, oral antidiabetic drugs

## Abstract

The discovery of new oral antidiabetic drugs remains a priority in medicine. This research aimed to evaluate the activity of the flavonoid baicalein and its natural glucuronide baicalin, compared to the antidiabetic drug metformin, as potential antiglycation, anti–radical, and anti-α–glucosidase agents, in order to assess their potential role in counteracting hyperglycemia-induced tissue damage. The study considered: (i) the BSA assay, to detect the formation of advanced glycation end products (AGEs), (ii) the GK peptide–ribose assay, which evaluates the cross–linking between the peptide and ribose, and (iii) the carbonyl content assay to detect the total carbonyl content, as a biomarker of tissue damage. In addition, to obtain a reliable picture of the antiglycation capacity of the investigated compounds, DPPH scavenging and oxygen radical absorbance capacity (ORAC) assays were performed. Furthermore, the anti–α–glucosidase activity of baicalein and baicalin was detected. Furthermore, to estimate cell permeability, preliminarily, the cytotoxicity of baicalein and baicalin was evaluated in HT–29 human colon adenocarcinoma cells using the MTT assay. Successively, the ability of the compounds to pass through the cytoplasmic membranes of HT–29 cells was detected as a permeability screen to predict in vivo absorption, showing that baicalein passes into cells even if it is quickly modified in various metabolites, being its main derivative baicalin. Otherwise, baicalin per se did not pass through cell membranes. Data show that baicalein is the most active compound in reducing glycation, α-glucosidase activity, and free radicals, while baicalin exhibited similar activities, but did not inhibit the enzyme α–glucosidase.

## 1. Introduction

Diabetes mellitus (DM) is the fourth leading cause of death in developed countries and is an epidemic in many developing countries. Numerous factors are involved in the aetiology of type 2 diabetes mellitus (T2DM), such as genetic predisposition, lifestyle, and dyslipidemia [1,2]. In poorly controlled DM, chronic hyperglycemia accelerates the formation of advanced glycation end products (AGEs) and the overproduction of reactive oxygen species (ROS) [3,4,5,6], increasing oxidative stress and glycation [7]. Furthermore, AGEs are ingested through food and then absorbed into the organism [8]. Several investigations have shown that AGEs and ROS are involved in a wide spectrum of pathologies, including cardiovascular failure, rheumatoid arthritis, Alzheimer’s disease, and kidney disease [9,10,11]. Therefore, inhibition of glycation reactions and free radicals may be an effective approach to counteract diabetes-associated diseases [12].

α–Glucosidase plays a key role in the regulation of blood sugar, and its inhibition reduces glucose absorption from food carbohydrates, reducing the postprandial glycemic peak [13]. Several authors have reported that acarbose, a well–known drug inhibitor of α–glucosidase, improves glycaemia and increases the level of GLP–1 (glucagon–like peptide–1) [13,14]. However, acarbose treatment causes various gastrointestinal disturbances that may not be produced by plant–derived inhibitors [15,16]. Furthermore, the discovery of new antihyperglycemic agents with multiple targets capable of reducing glucose–induced damage, decreasing glycation, ROS, and α–glucosidase activity could be relevant for patients with DM.

Baicalein (5,6,7–trihydroxyflavone) and baicalin (7–β–d–glucuronic acid of 5,6–trihydroxyflavone, Figure 1) have been identified in various medicinal plants, mainly in the roots of *Scutellaria baicalensis* G. (also known as the Chinese skull cap or Huang-Qin) and in the bark of *Oroxylum indicum* [17]. *Scutellaria baicalensis* is one of the 50 fundamental herbs of traditional Chinese medicine, widely used as an anti–inflammatory, antiviral, antibacterial, and anticancer remedy [18,19]. Studies have reported that baicalein and baicalin have antioxidant, anti–inflammatory, immune–stimulating, and antiviral activities [19,20,21]. Baicalin could also act in various intestinal diseases, as it can suppress the PI3K/AKT signalling pathway to inhibit the production of Bcl–2 proteins [22,23]. Previous studies have shown that baicalein and baicalin exhibit potential antidiabetic activities [24]. In detail, baicalein reduces oxidative stress, expressions of iNOS and TGF-β1, and NF–κB activation [25,26]. Furthermore, in both insulin deficiency and insulin–resistant rat models, baicalein showed antiglycation and anti–inflammatory protective mechanisms by also reducing AGE formation [27]. However, other authors did not find inhibition of protein glycation induced by 50 mM d-fructose [28]. Thus, this point may require further investigation.

Metformin (1,1–dimethylbiguanide) is a first–line oral antihyperglycemic agent largely used in T2DM due to its ability to decrease liver glucose production and improve insulin activity in target tissues [29]. Therefore, metformin reduces glycaemia and increases glyoxalase I activity, thus decreasing the methylglyoxal formation, a well–known precursor of AGEs [30,31]. Furthermore, the authors reported the in vitro inhibitory effect of metformin on protein glycation through various mechanisms [32,33,34]. 

This research aimed to investigate by a series of in vitro assays the activity of baicalein and its natural glucuronide baicalin, compared to metformin, as potential antiglycation agents and antiradical agents, in order to assess any possible role in counteracting tissue damage linked to diabetes. Furthermore, the occurrence of specific hypoglycemic effects was evaluated by investigating their anti–α–glucosidase activity. To complete the evaluation of flavonoids also under a predictive pharmacokinetic aspect, the ability of baicalein and baicalin to pass through HT–29 cell membranes was studied to estimate their bioavailability. 

## 2. Results and Discussion

### 2.1. Proteine Glycation Inhibition

Glycated albumin could be considered a diagnostic indicator for DM and has been proposed as a marker to evaluate atherosclerosis and coronary disease in diabetic patients [31]. To evaluate the capacity of the flavonoids baicalein and its natural 7–*O*–glucuronide baicalin, and the drug metformin to inhibit protein glycation and fluorescent AGE formation, three assays were used to evaluate different steps of glycation reactions (Appendix A). First, the BSA assay was performed to evaluate the ability of the selected compounds to reduce the AGE formation by the reaction between albumin and glycation agents. Second, the GK peptide–ribose assay was performed to estimate the ability of the compounds to inhibit peptide cross–linking; and third, the carbonyl content assay was performed, which measures the binding of DNP to protein carbonyls to detect the total amount of carbonyl–derived compounds in BSA–ribose glycation.

#### 2.1.1. BSA Assay

Glucose, glyoxal, and ribose were used as glycation inducers to detect the formation of AGEs in vitro. A nine-day course assay was performed showing that glucose, glyoxal, and ribose have different glycation capacities (Figure 2).

Ribose was a powerful glycation agent with maximal AGE formation after 5 days of incubation with BSA (Figure 2A). Glyoxal induced maximal glycation after seven days of incubation, while glucose was a very weak glycation inducer, with a slight increase in fluorescence after seven-nine days (Figure 2A). Therefore, the experimental protocols were set at five days for ribose and seven days for glyoxal and glucose; mean maximal glycation values are reported in Figure 2B. Under these different conditions, the ability of baicalein, baicalin, and metformin to reduce the formation of AGEs was studied (Figure 3).

In the glucose–BSA glycation test, both baicalein and baicalin significantly inhibited AGE formation from the concentration of 5 µM, while metformin did not show any significant inhibition (Figure 3A). Glyoxal-induced BSA glycation was inhibited by 10 µM baicalein and 5 µM baicalin (Figure 3B). Finally, with ribose, which causes the highest glycation, both baicalein and baicalin maintained their ability to reduce glycated albumin from the concentration of 5 µM, whereas metformin, as in the previous tests, did not achieve any protection (Figure 3C). Previous research suggested that metformin could be a moderate inhibitor of glycation, possibly due to its interaction with dicarbonyl compounds generated during the glycation process [34]. However, the concentration used in the reported in vitro study was elevated, as well as 1 mM, which is a very higher concentration, higher than the therapeutic level of the drug [33]. In fact, the therapeutic range is minor at 1.5 µM (max 5 µM), even if rarely, higher concentrations have been detected [35,36]. In our experimental condition, metformin (1–100 µM) did not show a significant effect even if a slight inhibition trend was observed (Figure 3).

#### 2.1.2. GK Peptide–Ribose Assay

This investigation was performed to evaluate the ability of the compounds to reduce peptide cross–linking, a process that occurs in the last phase of glycation. In the beginning, GK glycation was detected each day for up to 4 days, showing that AGE formation reached its maximum after 2 days (Appendix A). Therefore, the assay was performed after 2 days of incubation. It should be noted that baicalein (1–100 µM) inhibited AGE formation in a concentration–dependent manner (Figure 4). The IC_50_ was 49.7 ± 0.5 µM (pD_2_ = 4.3). Interestingly, its glucuronide derivative baicalin, tested in the same baicalein concentration range, only slightly and not significantly reduced the cross–linking of the GK peptide (Figure 4). As expected, metformin was not active as a linking inhibitor, in agreement with the results obtained with the BSA assay (Figure 4). 

#### 2.1.3. Carbonyl–Protein Content Assay

The generation of ROS during glycation and glycoxidation is capable of oxidizing the side chains of amino acid residues in protein to form carbonyl–protein derivatives, which are well–known to mediate free radical–induced damage to various biological macromolecules [37]. Therefore, the carbonyl content is considered an indicator of glycation and is used as a parameter to evaluate protein oxidation [38]. In this experimental step, metformin was not investigated because its antiglycation activity was very low in previous tests. Both baicalein and baicalin exhibited a detectable ability to reduce carbonyl increase in the concentration range of 10–100 µM, and baicalin also inhibited carbonyl formation at the lowest concentration tested of 5 µM. However, the inhibitory effects of both compounds were not concentration dependent, resulting in approximately 15–25% (Figure 5). Recently, some authors suggested that a sage methanol extract (0.25–1.00 mg/mL) significantly decreased BSA carbonyl formation [39], while other authors reported that single compounds, such as rosmarinic acid and carnosic acid (6.25–400 µM), decreased carbonylation [38]. Previously, various flavonoids, including baicalein, have been reported to decrease protein carbonylation in goat eye lenses [40].

### 2.2. Free Radical Inhibition

Hyperglycemia and ROS increase the formation of AGEs through oxidative steps (glycoxidation) [41]. For the evaluation of the antioxidant activity of the compounds under examination, two assays were used: (i) the DPPH assay, for which the reduction of the reference radical occurs by the transfer of a single electron to the radical molecule (single electron transfer mechanism, SET) [42]; and (ii) the ORAC assay, for which the compounds examined oxidize by donation of a hydrogen atom to radical molecules (hydrogen atom transfer mechanism (HAT) [43].

#### 2.2.1. DPPH^•^ Assay

Figure 6A shows the antiradical activity of the two flavonoids and metformin tested in the range of 0.1 to 500 µM. Ascorbic acid and *N*–acetylcysteine (NAC) were used as positive controls. Both baicalein and baicalin showed an antioxidant capacity comparable to that of the standard ascorbic acid, showing excellent scavenger activity. These compounds reach their maximum effect at the highest concentrations (≥50 µM), while NAC showed a lower potency, and metformin did not show any scavenging effect. Based on antiradical profiles, it was possible to quantify antioxidant efficacy by calculating the EC_50_ values, a parameter that indicates the effective substrate concentration that causes 50% of the maximum antioxidant response [44]. Potency values expressed as pD_2_ (−log EC_50_) are 4.91 ± 0.01 for baicalein and 4.89 ± 0.03 for baicalin, comparable to those of ascorbic acid (4.77 ± 0.04), and higher than those of NAC (4.38 ± 0.02, *p* < 0.05).

#### 2.2.2. ORAC Assay

Based on a different antiradical mechanism such as peroxyl radical scavenging capacity, the ORAC assay was performed [45]. Baicalein and baicalin antiradical activities were very similar and approximately ten times higher than metformin; the TEAC values were 5.41 ± 0.33, 5.90 ± 0.23, and 0.48 ± 0.10 µmol TE/µmol, respectively (Figure 6B). Ascorbic acid and NAC, standard antioxidants (0.65 ± 0.01 and 1.45 ± 0.01 µmol TE/µmol), showed lower activity than both flavonoids. The order of antioxidant activity was baicalin ≥ baicalein >> NAC > ascorbic acid ≥ metformin.

The results obtained with DPPH^•^ and ORAC assays agree with those of Gao et al. who suggested that flavonoids with o–dihydroxyl structure in the A ring are effective radical scavengers [20]. These can reduce the formation of AGEs by scavenging free radicals produced in the early phase of the glycation process [46]. Thus, the quenching of ROS by antioxidants such as baicalein and baicalin can contribute to the reduction of AGE production.

### 2.3. Anti-α–glucosidase Activity

The inhibitory activity of baicalein and baicalin was studied using the α–glucosidase assay, with *p*NPG as substrate and acarbose as positive control [47,48]. The absorbance curves were detected during the 60 min reaction between enzyme (0.05 µM) and *p*NPG (2 mM) alone and in the presence of different concentrations of baicalein (1–100 µM), revealing a concentration-dependent inhibition of enzyme activity (Figure 7A). Interestingly, under the same conditions, baicalin did not induce any modification of the α–glucosidase activity (Appendix A), suggesting that only the flavonoid aglycone could interact with the enzyme catalytic task. This observation is consistent with a previous study showing for baicalin an IC_50_ greater than 400 µM against rat α–glucosidase in vitro [49]. Therefore, an in–depth study on the enzyme was performed only with baicalein, as a potential new α–glucosidase inhibitor. Four α–glucosidase concentrations (0.035, 0.05, 0.07 and 0.1 µM) were tested to study the type of inhibition induced by baicalein (Appendix A and Figure 7B). 

Always, baicalein exhibited concentration–dependent inhibition, even with lower potency at higher enzyme concentrations (Figure 7B,C). Thus, the baicalein IC_50_ values increased at higher enzyme concentrations (Table 1). Baicalein 50 µM inhibited 0.035 and 0.05 µM α–glucosidase activity of about 90% and 75%, respectively.

To study the type of interaction between α–glucosidase and baicalein, the plot of enzyme activity (*v*) versus α–glucosidase enzyme concentration was obtained (Figure 7C), suggesting a concentration–dependent reversible interaction since all straight lines obtained at different concentrations of baicalein pass through the origin of the axes, and their slope decreased with increasing concentrations of the inhibitor. Previous research showed similar behavior for other natural compounds, for example, luteolin, morin, magnolol, and α–mangostin [48,50,51,52]. Furthermore, the type of baicalein inhibition was also estimated with Michaelis–Menten and Lineweaver–Burk plots (Figure 8A,B) obtained using different substrate concentrations (0.5, 1.0, 1.5 and 2.0 mM *p*NPG), while maintaining constant enzyme concentration (0.05 µM). The Lineweaver–Burk plot shows the intersection of lines in the second quadrant, as a result of the change in both V_max_ and K_m_ at different concentrations of baicalein, suggesting a mixed–type (mixture of competitive and non–competitive) mechanism of enzyme inhibition. When applying a mixed–model inhibition, V_max_ was 0.0926 ∆OD/min, K_m_ was 0.24 mM, and the baicalein inhibition constant (K_i_) was 25.89 µM. The baicalein K_i_ is very similar to those reported by other authors: 45 µM [53], 44 µM [54], and 14.6 µM [49].

### 2.4. Cellular Uptake 

The cytotoxicity assay was performed to see whether flavonoids could modify cell proliferation in culture and to determine the baicalein concentrations that would be useful to perform the uptake assay. The human colorectal adenocarcinoma (HT–29) cell line is a suitable model for bioavailability measurements thanks to its similarities, both in phenotype and in enzyme expression, with mature intestinal cells such as enterocytes [55]. Baicalein was tested from 0.1 to 50 µM after 24 h of incubation, displaying a slight inhibition of HT–29 cell viability; pD_2_ = 4.44 ± 0.09, showing low cytotoxicity (Appendix A).

#### Cellular Uptake of Baicalein in HT–29 Cells

The HT–29 cell line was used to study the ability of baicalein and baicalin to pass through the cellular membrane and therefore permeate into cells, estimating their bioavailability. The amount of compounds at extracellular and intracellular levels was detected by high–performance liquid chromatography (HPLC), equipped with a UV diode array detector. Preliminarily, cellular uptake was estimated by adding baicalein in a concentration range from 1 to 50 µM to the cell medium. Unexpectedly, after 30 min–3 h of incubation, it was observed that baicalein was no longer measurable and only three metabolites were detectable. In detail, the major component detected was the glucuronide derivative baicalin (Figure 9: peak 1, baicalin), both at the extra– and intracellular levels. Figure 10A shows that intracellular uptake increased with increasing baicalein concentrations in cell medium (3 h of incubation). Based on these data, subsequent tests were performed using baicalein at 5 µM, a concentration with low inhibitory activity on cell viability (−25% of the control).

After incubation with 5 µM baicalein, the glucuronide baicalin was identified both at the intracellular level: 0.60 ± 0.20 nmol/mg protein and at the extracellular level: 0.91 ± 0.24 µM. The results showed that the flavonoid baicalein is capable of passing through the HT–29 cell membranes and can also undergo a biotransformation process that leads to various metabolites (Figure 9 and Figure 10). The main metabolite has been identified as baicalein–7–glucuronide, the conjugated form of baicalein (phase II metabolism) that corresponds to baicalin. Otherwise, 5 µM baicalin was only slightly taken up in the intracellular compartment (0.069 ± 0.001 nmol/mg protein, Figure 10B). Baicalin remained for the most part in the extracellular compartment (3.02 ± 0.29 µM), showing a low ability to cross cell membranes. These data agree with various types of research that have indicated the conspicuous intestinal and liver metabolism of baicalein into baicalin through UDP–glucuronosyltransferases and the low bioavailability of baicalin [56,57,58,59].

## 3. Materials and Methods

### 3.1. Reagents

Acarbose, 2,2′–azobis(2–amidinopropane)–dihydrochloride (AAPH), baicalein (5,6,7–trihydroxyflavone), baicalin (7–β–d–glucuronic acid of 5,6–trihydroxyflavone), bovine serum albumin (BSA), 3–[4,5–dimethylthiazol–2–yl]–2,5 diphenyl tetrazolium bromide (MTT), 1,1–diphenyl–2–picrylhydrazil (DPPH^•^), α–glucosidase (EC 3.2.1.20, *Saccharomyces cerevisiae* type I, 10 U/mg protein), 6–hydroxy–2,5,7,8–tetramethylchroman–2–carboxylic acid (trolox), metformin, *p*–nitrophenyl–α–d–glucopyranoside (*p*NPG) and, generally, all chemicals and solvents were purchased from Merck KGaA, Darmstadt, Germany. Otherwise, the *N*–acetyl–glycyl–lysine methyl ester (GK) peptide was obtained from Bachem (Bachem AG, Bubendorf, Switzerland). The purity of the reference standards was 97%, while other chemicals were of at least analytical grade.

### 3.2. Protein Glycation Inhibition

#### 3.2.1. BSA Assay

Glycated BSA was obtained according to a previously reported method [48,60]. Briefly, AGEs were detected using BSA (50 mg/mL, pH 7.4), as protein substrate, and glucose (0.8 M), glyoxal (0.003 M) or ribose (0.1 M), as glycation agents. Each inhibitor was tested in the range of concentrations of 1*–*100 µM by adding to substrate solutions and incubation at 37 °C for a time of 5 (ribose) or 7 (glucose and glyoxal) days. The fluorescence intensity was measured at an excitation wavelength of 355 nm and an emission wavelength of 460 nm with a PerkinElmer Victor Nivo microplate reader (Waltham, MA, USA). Inhibition of AGE formation was calculated as the fluorescence difference between glycation under the control condition and in the presence of each inhibitor. Aminoguanidine (2.5 mM, AG) was used as a positive control [61].

#### 3.2.2. GK Peptide–ribose Assay

The *N*–acetyl–glycyl–lysine methyl ester (GK) peptide–ribose assay was performed with some modifications to a published method [62]. Briefly, GK peptide (27 mg/mL), a synthetic model peptide, was incubated with 40 mg/mL (0.27 M) d–ribose in sodium phosphate buffer (pH 7.4) and test samples were added at final concentrations ranging from 1.0–100 μM. After 48 h of incubation, fluorescence was read at 355 and 460 nm for excitation and emission wavelengths, respectively, with a PerkinElmer Victor Nivo microplate reader (Waltham, MA, USA). The fluorescence measurement allows the calculation of GK peptide glycation inhibition as a percentage difference between the control condition and glycation in the presence of the inhibitor. Aminoguanidine (2.5 mM, AG) was used as a positive control.

#### 3.2.3. Carbonyl Content Assay

The total protein–bound carbonyl content was determined by derivatization of the carbonyl group with 2,4–dinitrophenylhydrazine (DNPH), which leads to the formation of a stable dinitrophenylhydrazone product [63], which was monitored spectrophotometrically at 375 nm with a PerkinElmer Victor Nivo microplate reader (Waltham, MA, USA). Glycated BSA samples were mixed with DNPH solution (10 mM in 2.0 M HCl) for 1 h at 37 °C in the dark. The proteins were precipitated with 20% trichloroacetic acid. The mixture was centrifuged at 10,000× *g* rpm, 10 min at 4 °C, to obtain a pellet and the supernatant was removed. DNPH was removed by extracting the pellet twice using a solution of ethyl acetate/ethanol (1:1). Lastly, the pellet was dissolved in guanidine hydrochloride (6 M) before reading the absorbance at 375 nm. The concentration of DNPH derivatized proteins was determined by the molar extinction coefficient of 22,000 M^−1^ cm^−1^ [64]. The protein concentration was measured using a BCA assay [65], and the protein carbonyl concentration was expressed as nmol/mg of protein. Aminoguanidine (2.5 mM, AG) was used as a positive control.

### 3.3. Free Radical Inhibition

#### 3.3.1. DPPH^•^ Assay

The free radical scavenging activity of the samples was measured using the DPPH^•^ method [66]. Sample solutions were prepared and added to 70 µM DPPH^•^; the mixtures were kept in the dark for 60 min and absorbance was read at 517 nm using a Beckman Coulter DU 8005 spectrophotometer (Fullerton, CA, USA). The radical scavenging capacity was expressed as a percentage effect.

#### 3.3.2. ORAC Assay

The radical oxygen absorption capacity (ORAC) assay was performed as previously described [67]. Briefly, trolox was prepared in phosphate buffer over a concentration range of 6.25 to 50 µM. In 24–well plates, 1.5 mL of fluorescein (0.08 µM) was added, followed by 250 µL of trolox for the control, 250 µL of buffer for the blank, and 250 µL of each inhibitor (5.0 µM). After 10 min of incubation, at 37 °C, 250 µL of 0.15 M AAPH was added. Successively, the PerkinElmer Victor Nivo microplate reader (Waltham, MA, USA) was settled for a fluorescence kinetic reading at 37 °C for one hour, with excitation and emission wavelengths, respectively, of 485 and 530 nm. The results were expressed in TEAC (trolox equivalent antioxidant capacity, TE µmol/µmol compound).

### 3.4. α– Glucosidase Inhibition

The inhibitory activity of baicalein and baicalin was studied using the yeast α–glucosidase assay, using *p*NPG as a substrate. The enzyme acts by hydrolyzing *p*NPG into α–d–glucopyranoside and *p*–nitrophenol (yellow); the reaction was detected at different inhibitor concentrations by variation in chromatic intensity [47,48]. Acarbose (1.25 M) was used as a positive control. Each sample was incubated with several α–glucosidase concentrations (0.035, 0.05, 0.07 and 0.1 µM) in PBS solutions (pH 6.8) for 10 min, at 37 °C. The reaction started with the addition of *p*NPG. Absorbance values were detected at 405 nm for 60 min, using a PerkinElmer Victor Nivo microplate reader spectrophotometer (Waltham, MA, USA). The half–maximal inhibitory concentration (IC_50_) was estimated by the plot of relative enzyme activity *versus* inhibitor concentration. The type of enzyme inhibition exerted by baicalein was evaluated from kinetic studies using different substrate concentrations (0.5, 1.0, 1,5 and 2.0 mM *p*NPG) through the Michaelis–Menten and Lineweaver–Burk plots.

### 3.5. Cellular Uptake

#### 3.5.1. MTT Assay

Cell viability was assessed with the MTT assay [46]. HT–29 cells were seeded in a 96–well plate at a density of 5000 cells in each well and allowed to grow overnight. The cells were treated with various concentrations of the substances. After 24 h of incubation, an aliquot of MTT solution was added to each well, to reach the final concentration of 500 μg/mL. Subsequently, after the MTT reduction by cellular enzymes, the medium was removed and the insoluble formazan salts were solubilized with 2–propanol. The absorbance of each purple formazan solution was measured using a PerkinElmer Victor Nivo microplate reader (Waltham, MA, USA), at a wavelength of 520 nm.

#### 3.5.2. HT–29 Uptake Assay

To evaluate the cellular uptake of compounds, a previously described method was used [68]. Briefly, HT–29 cells were seeded in a 6–well plate at a density of 500,000 cells in each well in a complete medium and allowed to grow until confluence for 48 h, at 37 °C (Figure 11). The culture medium was then removed and the cells were treated with each compound (5 µM). After incubation, an aliquot of the extracellular solution was taken from each well and stored at −20 °C for further analysis. The medium was then removed, the cells were washed with PBS and prepared for the following steps: (i) protein quantification, cells were lysed with 200 μL of lysis buffer and the wells were washed with 200 μL of PBS; and (ii) evaluation of cell uptake, cells were gently scraped with PBS, centrifuged at 1250× *g* rpm for 5 min, and added with 500 μL of 80% *v*/*v* methanolic solution in water/0,1% *v*/*v* acetic acid to extract intracellular content. The samples were kept on ice for 15 min, then sonicated and centrifuged at 20,000× *g* rpm for 10 min. Precipitated proteins were excluded, while supernatant solutions were collected for chromatographic analysis. The lysates and intracellular solutions were stored at −20 °C for HPLC–DAD analysis (Figure 11). The peaks were identified by comparing the chromatogram relative to the intracellular and extracellular contents with the chromatogram of each analytical standard.

#### 3.5.3. High-Performance Liquid Chromatography (HPLC–DAD) Analysis

Chromatographic analyses were performed with an HPLC instrument (Waters Corporation, Milford, DE, USA) equipped with a 1525 binary HPLC pump and a 2998 photodiode array detector. A Symmetry^®^ RP C18 column, 4.6 × 75 mm, 3.5 μm column (Waters, Milford, USA) was used. As previously reported [69], a linear methanolic gradient was applied from 20 to 80% *v*/*v* in 12 min (A: water with 0.1% *v*/*v* acetic acid, and B: methanol with 0.1% *v*/*v* acetic acid), with a flow rate of 1 mL/min. Peaks were detected in the range of 210–400 nm. All samples were filtered with membrane filters (0.22 μm pore size, Millipore, Merck KGaA, Darmstadt, Germany) and then injected (10 μL). Standard stock solutions were prepared in methanol. The amount of each compound was calculated using a standard calibration curve. The protein content of the lysate was detected using the Lowry protein assay [70], to express the results as nmol/mg protein.

### 3.6. Statistical Analysis

Data are expressed as mean ± SEM of at least three independent experiments. The sigmoid curve fitting and statistical evaluations were performed using GraphPad Prism 9 (San Diego, CA, USA). Statistical comparisons among three or more groups were performed using one–way ANOVA, followed by Dunnett’s multiple comparison test or Tukey’s test. The level of significance was established at *p* < 0.05.

## 4. Conclusions

This research shows that both baicalein and baicalin have appreciable antiglycation and antiradical activities comparable to each other. On the contrary, the drug metformin shows only modest antiradical activity, without the capacity to decrease glycation reactions. Furthermore, baicalein, but not baicalin, inhibits α–glucosidase activity. Therefore, the natural compound baicalein exhibits antiglycation, antioxidant and anti–α–glucosidase activities, while baicalin does not cause any enzyme inhibition. Baicalein exhibits a mixed–type reversible inhibition on α–glucosidase, showing a K_i_ of 25.89 µM. Natural compounds baicalein and baicalin show favorable antidiabetic properties, supporting their intake through diets or even plant-based preparations.

In particular, the predicted favorable absorption profile of baicalein makes this natural flavonoid a promising candidate for further investigations about the possibility of developing a formulation of baicalein, finding the appropriate dose and pharmaceutical preparation, to be used in the prevention of hyperglycemia–related diseases.

## Figures and Tables

**Figure 1 pharmaceutics-14-02141-f001:**
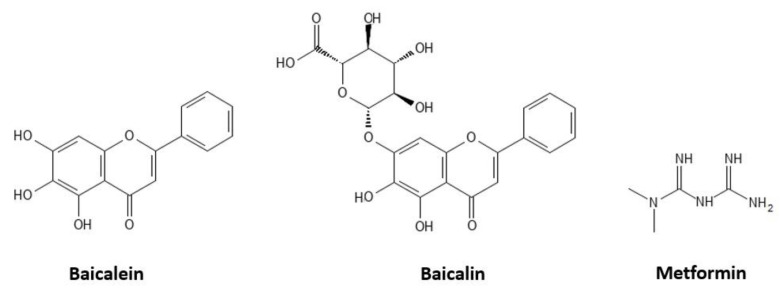
Chemical structures of baicalein, baicalin, and metformin.

**Figure 2 pharmaceutics-14-02141-f002:**
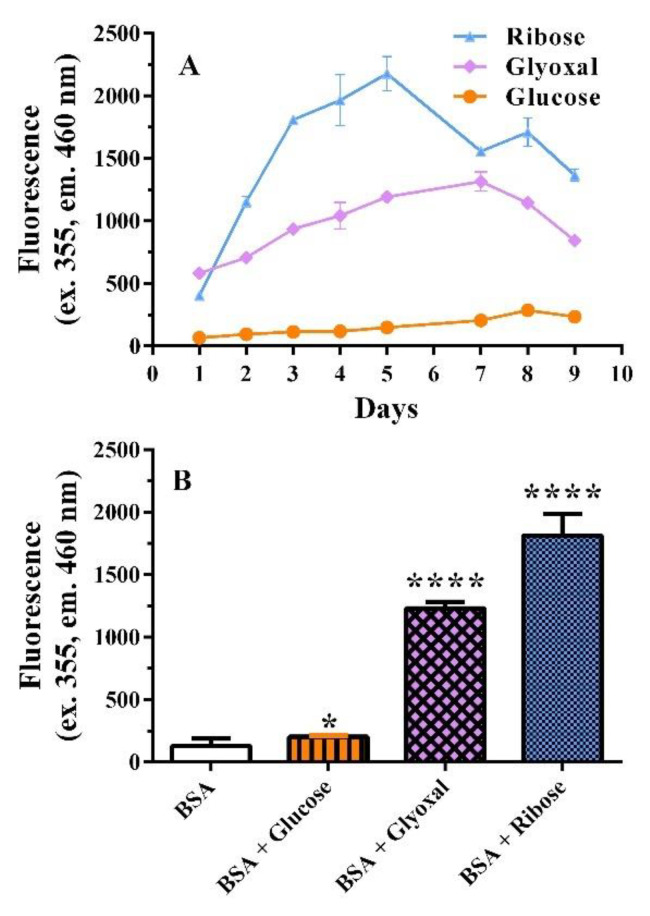
(**A**) Time course of AGE fluorescence during incubation of 50 mg/mL BSA with 0.1 M ribose, 0.003 M glyoxal and 0.8 M glucose. (**B**) Maximal AGE formation after 5 (ribose) and 7 days of incubation with the selected glycation agent. Data are the mean ± SEM of 3–6 experiments. * *p* < 0.05 and **** *p* < 0.0001 versus BSA (control). Turkey’s analysis test shows significant differences between BSA+glucose versus BSA+glyoxal or BSA+ribose (*p* < 0.0001), or BSA+glyoxal versus BSA+Ribose (*p* < 0.01).

**Figure 3 pharmaceutics-14-02141-f003:**
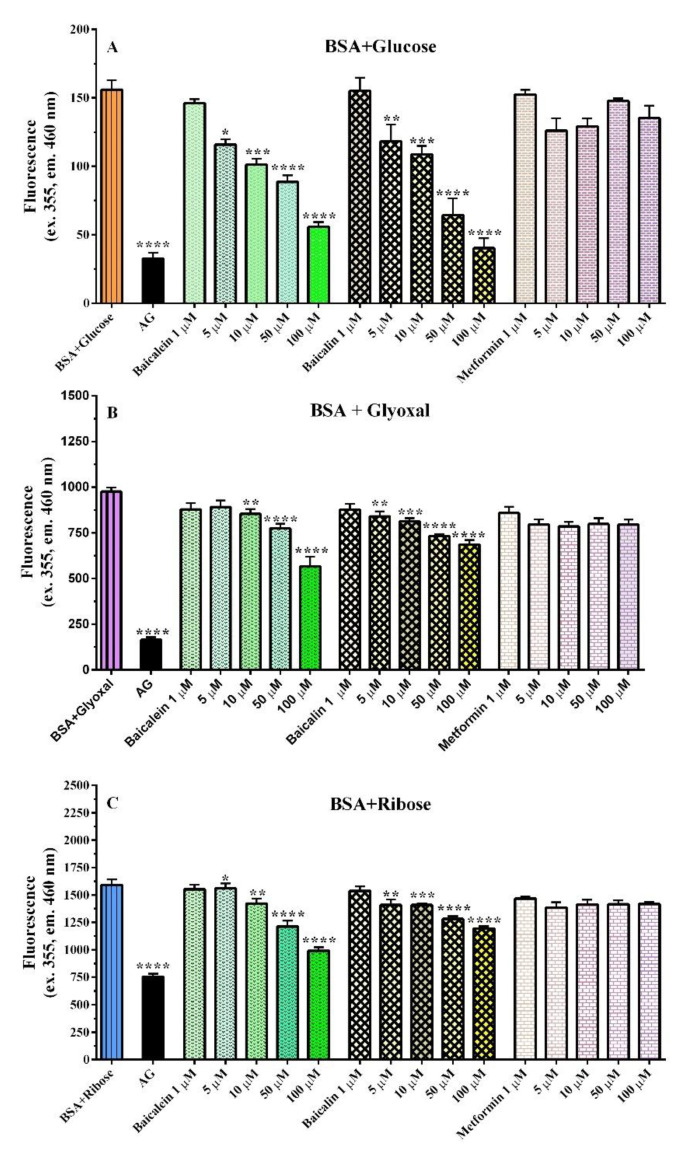
Baicalein (green bars), baicalin (yellow bars), and metformin (pink bars) inhibition of AGE formation after 7(**A**,**B**) or 5 (**C**) days of incubation of 50 mg/mL BSA with 0.8 M glucose (**A**), 0.003 M glyoxal (**B**), and 0.1 M ribose (**C**). Aminoguanidine (2.5 mM, AG) was the positive control. Data are the mean ± SEM of 3–6 experiments. * *p* < 0.05, *** *p* < 0.001, **** p* < 0.001, ***** p* < 0.0001 versus BSA glycation (BSA + glycation agent).

**Figure 4 pharmaceutics-14-02141-f004:**
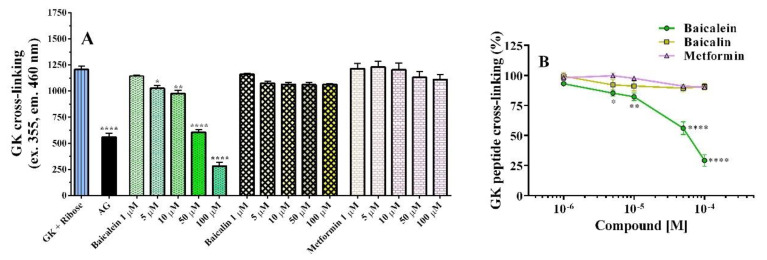
Effects of baicalein (green bars), baicalin (yellow bars), and metformin (pink bars) on GK cross–linking formation after 2 days of incubation of 26.7 mg/mL GK peptide (0.1 M) with 40 mg/mL ribose (0.27 M) reported as fluorescence (**A**) and percentage of control fluorescence (**B**). Aminoguanidine (2.5 mM, AG) was the positive control. Data are the mean ± SEM of 3–6 experiments. * *p* < 0.05, ** *p* < 0.01, and **** *p* < 0.0001 versus GK cross–linking (GK + Ribose).

**Figure 5 pharmaceutics-14-02141-f005:**
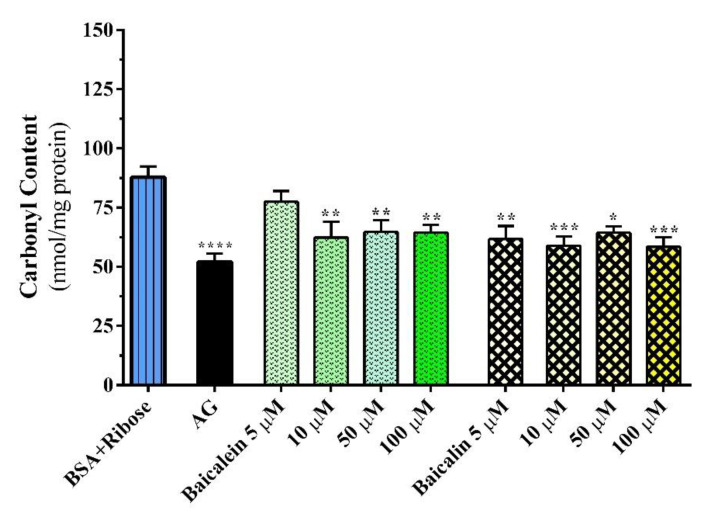
Effects of baicalein (green bars) and baicalin (yellow bars) on carbonyl content, detected with 50 mg/mL BSA and 0.1 M ribose, reported as carbonyl content and expressed in nmol/mg protein. Aminoguanidine (2.5 mM, AG) was the positive control. Data are the mean ± SEM of 3–6 experiments. * *p* < 0.05, ** *p* < 0.01, *** *p* < 0.001, and **** *p* < 0.0001 versus maximal carbonyl content (BSA + Ribose).

**Figure 6 pharmaceutics-14-02141-f006:**
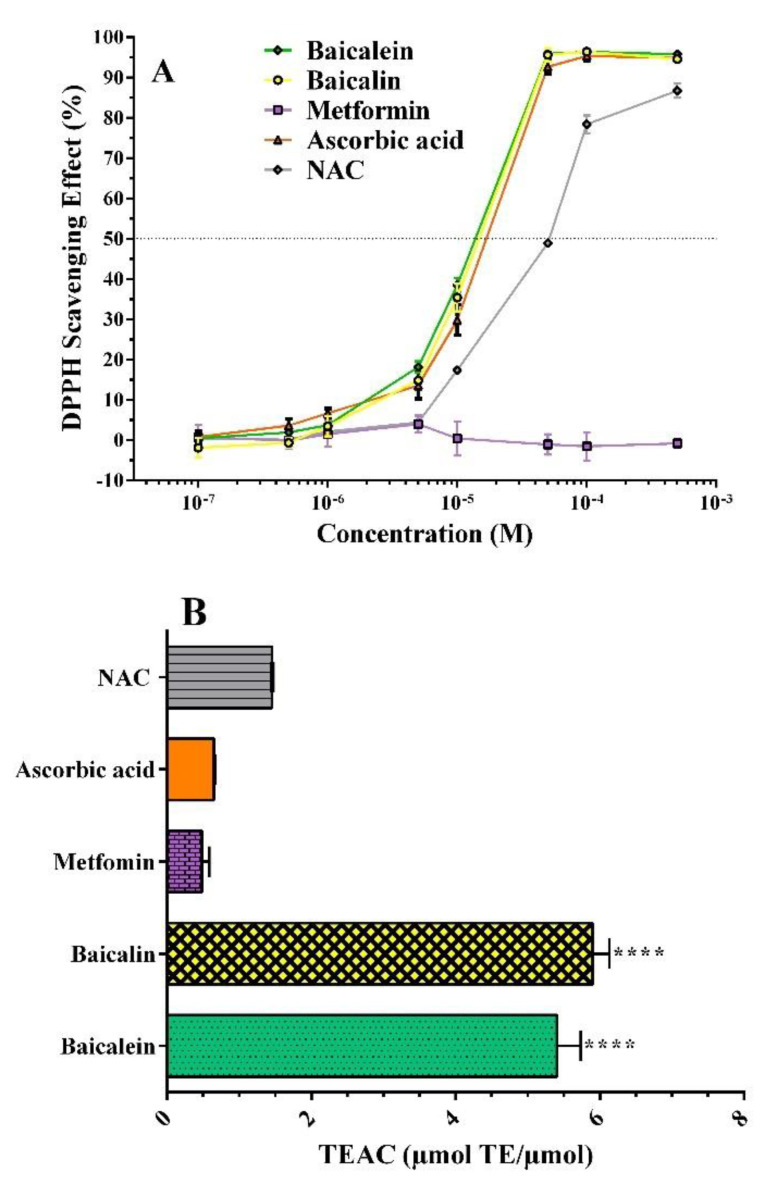
Antiradical activity of baicalein, baicalin, and metformin detected with DPPH^•^ (**A**) and ORAC (**B**) assays. Positive controls: ascorbic acid and *N*–acetylcysteine (NAC). Data are the mean ± SEM of 3–6 experiments. **** *p* < 0.0001 versus positive controls. ORAC values are expressed as TEAC (trolox equivalent antioxidant capacity, µmol TE/µmol compound).

**Figure 7 pharmaceutics-14-02141-f007:**
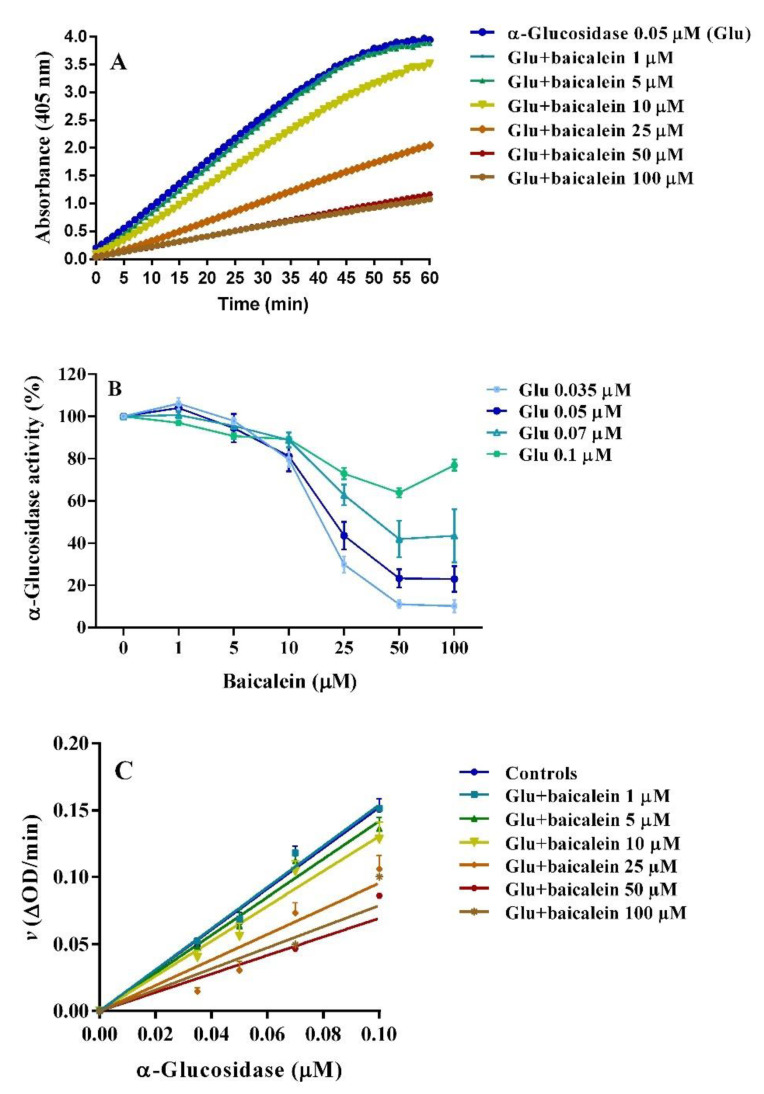
(**A**) An example of kinetic curves of 0.05 µM α–glucosidase, using 2 mM *p*NPG as substrate, in the presence of increasing concentrations of baicalein. (**B**) Inhibition of baicalein was studied using four different concentrations of α–glucosidase: 0.035, 0.05, 0.07 and 0.1 µM. (**C**) Plot of enzyme activity (*v*) versus α–glucosidase enzyme concentration. Data are reported as mean ± SEM of 3–6 experiments.

**Figure 8 pharmaceutics-14-02141-f008:**
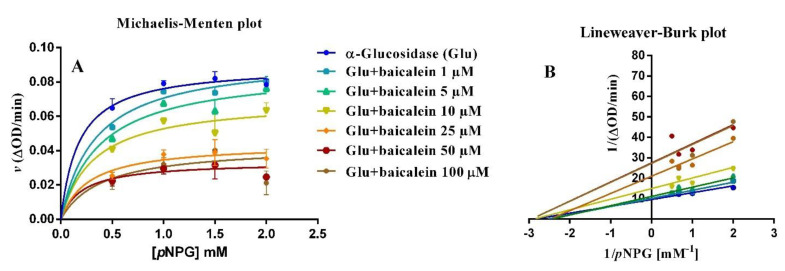
Michaelis–Menten (**A**) and Lineaweaver–Burk (**B**) graphs of α–glucosidase activity with different substrate concentrations (0.5–2.0 mM pNPG) in the presence of baicalein (1–100 µM). Data are obtained from 4–5 experiments.

**Figure 9 pharmaceutics-14-02141-f009:**
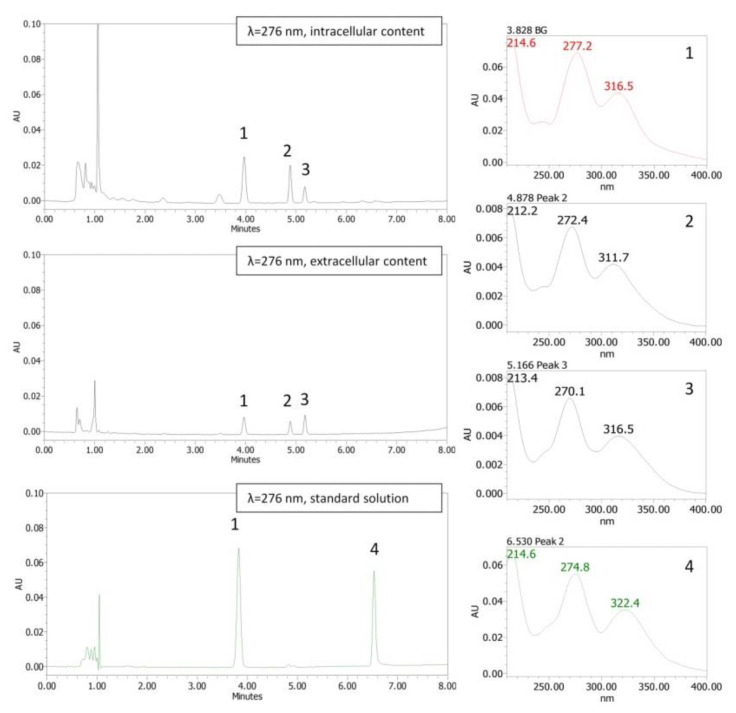
HPLC chromatographic analysis of compounds detected in HT–29 cells after 3 h of incubation of 5 µM baicalein. Chromatograms of the intracellular, extracellular, and standard contents are reported. The panels on the right show the UV spectra of the peaks revealed during the analysis. Peak 1 has been identified as baicalein-7-glucuronide (baicalin), while peaks 2 and 3 are unknown metabolites. Standard baicalein peak 4, Rt 6.53 min; baicalin standard, peak 1, Rt 3.83 min.

**Figure 10 pharmaceutics-14-02141-f010:**
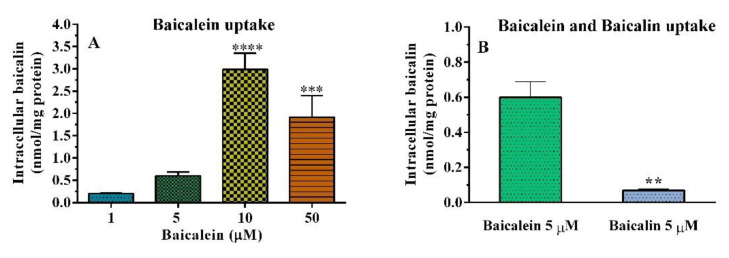
Intracellular baicalin content measured in HT–29 cells after 3 h of incubation with baicalein (1–50 µM, (**A**)), and baicalein and baicalin at 5 µM (**B**). Each value represents the mean ± SEM of at least 3 experiments. ** *p* < 0.01, *** *p* < 0.001 and **** *p* < 0.0001 versus intracellular baicalin content detected using baicalein 1 µM (**A**) or baicalein 5 µM (**B**).

**Figure 11 pharmaceutics-14-02141-f011:**
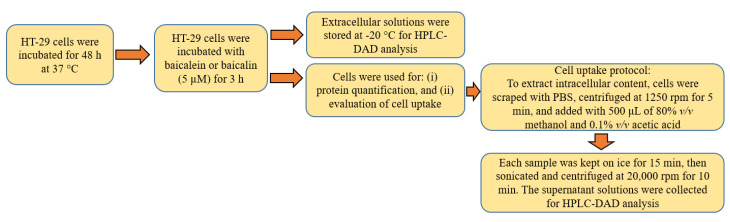
Flowchart of the experimental protocol used to assess the cell uptake of baicalein and baicalin.

**Table 1 pharmaceutics-14-02141-t001:** Inhibitory potency of baicalein with four different α–glucosidase concentrations.

Baicalein (IC_50_)		
α-Glucosidase	10^−5^ M	pD_2_
0.035	1.81	4.74 ± 0.02 ^a^
0.05	2.41	4.62 ± 0.04 ^a^
0.07	5.16	4.29 ± 0.07 ^a^
0.10	55.49	3.26 ± 0.26 ^b^

IC_50_: half maximal inhibitory concentration of enzyme activity. Data are obtained using the non-linear regression of the normalized response of 4–5 experiments. pD_2_: −Log IC_50_. ^a,b^ Different superscript letters indicate significant difference (*p* < 0.0001).

## Data Availability

All results are contained in the article and Appendix A.

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
