# Peer review of "Comparative Evaluation of the Antiglycation and Anti-α-Glucosidase Activities of Baicalein, Baicalin (Baicalein 7-O-Glucuronide) and the Antidiabetic Drug Metformin"

_pharmaceutics, 2022, doi:10.3390/pharmaceutics14102141_

Round 1

Reviewer 1 Report

Comments on manuscript “Comparative Evaluation of the Antiglycation and Anti-α-Glucosidase Activities of Baicalein, Baicalin (Baicalein 7-O-Glucuronide) and the Antidiabetic Drug Metformin

08/10/2022 Hsuan Ping Chang

In this manuscript, the author investigated the therapeutic potentials of two natural compounds, baicalein and baicalin, for the treatment of diabetes mellitus (DM). By conducting a series of in vitro assays, the authors found that baicalein enables the reduction of glycation, α-glucosidase activity, and free radicals. On the other hand, baicalein is capable of reducing glycation and free radicals but not α-glucosidase activity. Besides, by using cell-based assay, the author found that baicalein has the ability to pass the cell membrane and concluded that this compound has potential absorption in vivo. While the author shows positive results for these two natural compounds when compared to current standard DM treatment drugs, including metformin and acarbose, some major concerns need to be addressed before considering them appropriate for publication.

First, the sections of this manuscript may need to be modified. Based on the instruction from the journal, I believe it should be Introduction, Method, Result, and Discussion. Especially, the Result and Discussion should be in different sections. Moreover, it is expected that a manuscript will have a more detailed Discussion. Based on each finding in this manuscript, appropriate and more references should be provided for compare-and-contrast. Secondly, regarding the suitability of this manuscript to fit the submitted section “Pharmacokinetics and Pharmacodynamics”, it would be somehow difficult to relate the research conducted in this manuscript to the submitted section. Especially, one would expect the compounds investigated to be tested in an in vivo system. For example, there may have been published research that investigated glycation in mice, rats, or other species. Although the author did perform a cellular uptake assay and relate it to the bioavailability of baicalein, it may not be enough to indicate the pharmacokinetics and pharmacodynamics properties of the compound. Should the author provide some in vivo results, the manuscript may fit this section.

Third, it would not be easy to comprehensively evaluate the antiglycation capability of the investigated compounds based on the individual in vitro assays. It would be preferable if the author could provide figures/schemes outlining the pathway of AGE formation and how each in vitro assay correlated to the corresponding steps. Then, the author may consider mentioning steps or properties within the AGE pathway that were not tested and their contributions to the final evaluation of these two compounds.

Fourth, would the author indicate the study objective more clearly? The objective of the study should be mentioned in the last part of the Introduction and in the abstract. Since the study objective is not clearly defined, it is quite confusing why the author compared the investigated compounds with metformin and acarbose. In addition, would the author mention the innovation of the current study? For each result in this manuscript, it seems the author mainly provided references and commented on whether their finding corresponded to the previous reports. The knowledge added to the field should be highlighted.

Minor comments:

The data should be presented as mean and standard deviation (SD) instead of SEM.

Figure 10 is missing.

Figures 2 and 3: Please explain why the fluorescence values of BSA+glucose/glyoxal/ribose are different between Figures 2A and 3 (dashed line).

Figures 3A and 3B provided similar information and may seem duplicated.

Figure 6: Please describe the definition of “v versus

Table 1: the footnotes of a and b are missing. 

Author Response

Response to Reviewer 1

 The authors thank the reviewer for the valuable suggestions in order to improve the scientific merit of the manuscript. Here, we provide detailed answers to all the raised questions.

“In this manuscript, the author investigated the therapeutic potentials of two natural compounds, baicalein and baicalin, for the treatment of diabetes mellitus (DM). By conducting a series of in vitro assays, the authors found that baicalein enables the reduction of glycation, α-glucosidase activity, and free radicals. On the other hand, baicalein is capable of reducing glycation and free radicals but not α-glucosidase activity. Besides, by using cell-based assay, the author found that baicalein has the ability to pass the cell membrane and concluded that this compound has potential absorption in vivo. While the author shows positive results for these two natural compounds when compared to current standard DM treatment drugs, including metformin and acarbose, some major concerns need to be addressed before considering them appropriate for publication.

First, the sections of this manuscript may need to be modified. Based on the instruction from the Journal, I believe it should be Introduction, Method, Result, and Discussion. Especially, the Result and Discussion should be in different sections. Moreover, it is expected that a manuscript will have a more detailed Discussion. Based on each finding in this manuscript, appropriate and more references should be provided for compare-and-contrast. Secondly, regarding the suitability of this manuscript to fit the submitted section “Pharmacokinetics and Pharmacodynamics”, it would be somehow difficult to relate the research conducted in this manuscript to the submitted section. Especially, one would expect the compounds investigated to be tested in an in vivo system. For example, there may have been published research that investigated glycation in mice, rats, or other species. Although the author did perform a cellular uptake assay and relate it to the bioavailability of baicalein, it may not be enough to indicate the pharmacokinetics and pharmacodynamics properties of the compound. Should the author provide some in vivo results, the manuscript may fit this section.

  • The authors thank the reviewer for comments. Regarding the sections of the manuscript, the Journal’s instructions for authors indicate that the Discussion section “may be combined with Results”, (https://www.mdpi.com/journal/pharmaceutics/instructions); therefore, in order to make easier for the reader to follow the entire data flow reported in the manuscript, we chose this option allowed for Research Manuscript type.

Regarding the issue on the suitability of the manuscript to fit the section “Pharmacokinetics and Pharmacodynamics”, we believe that the plan of the in vitro experiments, considering the enzyme kinetics and cell uptake studies, can fit both the Pharmacokinetics and the Pharmacodynamics topics. It was not our aim to investigate in vivo properties of the compounds that could represent a future advancement of the research, based on the present results obtained in specific in vitro conditions. Furthermore, baicalein and baicalin are already used as components of medicinal plant extracts or food supplements, and thus the detection in simple pharmacokinetics models of cellular permeability could be useful to support the current use of these products.

Third, it would not be easy to comprehensively evaluate the antiglycation capability of the investigated compounds based on the individual in vitro assays. It would be preferable if the author could provide figures/schemes outlining the pathway of AGE formation and how each in vitro assay correlated to the corresponding steps. Then, the author may consider mentioning steps or properties within the AGE pathway that were not tested and their contributions to the final evaluation of these two compounds.

  • A scheme describing the pathway of AGE formation and the corresponding steps evaluated with each in vitro assay has been introduced as a supplementary figure (Figure S1).

Fourth, would the author indicate the study objective more clearly? The objective of the study should be mentioned in the last part of the Introduction and in the abstract. Since the study objective is not clearly defined, it is quite confusing why the author compared the investigated compounds with metformin and acarbose. In addition, would the author mention the innovation of the current study? For each result in this manuscript, it seems the author mainly provided references and commented on whether their finding corresponded to the previous reports. The knowledge added to the field should be highlighted.

  • A better explanation of the critical suggested points has been provided in the manuscript. As suggested, the purpose of this research has been declared in the last part of the Introduction and in the abstract. However, for the abstract the typographical rules of the Journal require a total of about 200 words maximum; thus, the abstract was only slightly changed.

Minor comments:

The data should be presented as mean and standard deviation (SD) instead of SEM.

  • We thank the reviewer for comment; however, we used standard error of the mean (SEM) as an inferential parameter of the sampling process in our experimental design. The aim was to give information about the precision of the data, and how well the sample can represent the entire population of data, since SEM provides a description of how far is the sample mean from the population mean; the use of SEM and respective inferential error bars (see Cumming et al., Journal of Cell Biology, 177 (1), 7-11, 2007) helps in comparing samples from two groups, to see if they are different. On the other hand, there is less interest here about the degree to which single measurements within the sample differ from the sample mean, as indicated by standard deviation (SD).

Figure 10 is missing.

- We are sorry for the mistake, due to a preliminary figure numbering. We fixed the problem in the revised version. Furthermore, the figures were updated graphically, improving their legibility.

Figures 2 and 3: Please explain why the fluorescence values of BSA+glucose/glyoxal/ribose are different between Figures 2A and 3 (dashed line).

  • The fluorescence values of BSA+glucose/glyoxal/ribose are different between Figures 2A and 3 because they are from different experiments; in a first phase, the maximal glycation was studied alone using different days of incubation. Successively, in a second phase, glycation experiments were performed using the inhibitors. In each set of experiments, we used their own controls. For this reason, the absolute fluorescence data are relatively dissimilar.

Figures 3A and 3B provided similar information and may seem duplicated.

  • Figure 3A (now 4A) shows the absolute values of GK cross-linking, together with the reference values (GK peptide + ribose and AG positive control), while Figure 3B (now 4B) reports the normalized percentage data. We believe that both panels can provide useful information.

Figure 6: Please describe the definition of “v versus”

  • (Now is Figure 7) We provided a detailed explanation of the term, that is, the enzyme activity (v) versus α–glucosidase enzyme concentration.

Table 1: the footnotes of a and b are missing.

- The superscript letters refer to the “superscript method” used in statistical analysis, which is a current way of expressing multiple pairwise comparisons (see, for example, discussion by H.P. Piepho doi:10.2134/agronj2017.10.0580). The letter display allows an efficient reporting method of statistical significance, that permits one to avoid the use of an excessive number of different symbols, as occurring with post hoc statistical tests of significance, that perform all the possible paired combinations, also considering the family-wise error rate. Therefore, the interpretation of these superscripts in Table 1 is, as reported in the footnotes: “Values not sharing a common superscript letter differ significantly at p< 0.0001.”

Reviewer 2 Report

The present manuscript entitled “Comparative Evaluation of the Antiglycation and Anti-α-Glucosidase Activities of Baicalein, Baicalin (Baicalein 7-O-Glucuronide) and the Antidiabetic Drug Metformin" from Froldi et al involved the interesting approach to analyze the bioactivity of two flavonoid compounds (baicalein and baicalin) in comparison with the current commercial drug metformin.

The introduction is comprehensive and according to the developed topic of the manuscript, and it has revised bibliographical references to support the research.

Also, the manuscript is interesting, clear, organize, and focused on the topic that is of increasing interest due to the potential applications of natural products in different areas of the sciences such as toxicology, biomedicine, pharmacology, etc.

Additionally, this investigation showed the continuous research the authors are developing to find the potential solution for an increasing health issue such as diabetes by using natural products.

In addition, it has an organize and appropriate methodology (extensively discussed) to analyze the potential antiglycation properties, the anti-α-glucoside activity, and the capability of them to pass through HT-29 cell membranes of these flavonoid compounds (protein glycation inhibition assay, BSA assay, GK peptide-ribose assay, carbonyl-protein content assay, free radical inhibition assays, anti- α-glucoside activity, cellular uptake, etc.).

Additionally, the data they reported is supported with clear and logical images/figures that summarize all the required data for the discussion item (and appropriate statistics analyses).

Besides, I encourage the authors to check some mistakes (in yellow, pdf attached) such as:

- Please remember to use italics in chemical nomenclature such as N-acetylcysteine, o-dihydroxyl, etc.

- Please homologate the size and the style of the graphics throughout the manuscript.

-Please improve the image quality of the figures and images throughout the manuscript (the words are blurry, and the graphics look pixelated).

Besides, I suggest improving the conclusions/future perspectives mentioning (venturing thoughts) which will be the possible clinical formulation (in the manuscript the authors mentioned plant-based preparations), dose or utilization of both natural products as antidiabetic agents (not only taking them through diets).

Finally, I would like to invite the authors to include the abbreviation list of words at the end of this manuscript.

I recommend the acceptance of this manuscript after the authors performed the suggested corrections/additions.

Author Response

Response to Reviewer 2

Thank you very much for your appreciated review that assisted us to improve the manuscript and also encouraged us in future researches. The suggestions have been taken into account, and all changes are reported with revision tracking in Word in the attached revised manuscript.

The present manuscript entitled “Comparative Evaluation of the Antiglycation and Anti-α-Glucosidase Activities of Baicalein, Baicalin (Baicalein 7-O-Glucuronide) and the Antidiabetic Drug Metformin" from Froldi et al. involved the interesting approach to analyze the bioactivity of two flavonoid compounds (baicalein and baicalin) in comparison with the current commercial drug metformin.

The introduction is comprehensive and according to the developed topic of the manuscript, and it has revised bibliographical references to support the research.

Also, the manuscript is interesting, clear, organize, and focused on the topic that is of increasing interest due to the potential applications of natural products in different areas of the sciences such as toxicology, biomedicine, pharmacology, etc.

Additionally, this investigation showed the continuous research the authors are developing to find the potential solution for an increasing health issue such as diabetes by using natural products.

In addition, it has an organize and appropriate methodology (extensively discussed) to analyze the potential antiglycation properties, the anti-α-glucoside activity, and the capability of them to pass through HT-29 cell membranes of these flavonoid compounds (protein glycation inhibition assay, BSA assay, GK peptide-ribose assay, carbonyl-protein content assay, free radical inhibition assays, anti- α-glucoside activity, cellular uptake, etc.).

Additionally, the data they reported is supported with clear and logical images/figures that summarize all the required data for the discussion item (and appropriate statistics analyses).

  • The authors thank the reviewer for favorable comments and suggestions to improve the scientific merit of the manuscript. All the suggested changes have been made.

Besides, I encourage the authors to check some mistakes (in yellow, pdf attached) such as:

- Please remember to use italics in chemical nomenclature such as N-acetylcysteine, o-dihydroxyl, etc.

- Thank you for the suggestion; the manuscript was carefully checked and the corrections made.

Please homologate the size and the style of the graphics throughout the manuscript.

  • Graphical improvements have been performed. All figures were updated to improve their graphical definition.

Please improve the image quality of the figures and images throughout the manuscript (the words are blurry, and the graphics look pixelated).

  • The blurry quality of words in the previous version was also a consequence of the quick conversion from the Word file. Now image quality has been improved.

Besides, I suggest improving the conclusions/future perspectives mentioning (venturing thoughts) which will be the possible clinical formulation (in the manuscript the authors mentioned plant-based preparations), dose or utilization of both natural products as antidiabetic agents (not only taking them through diets).

  • Recommended improvements have been included in the revised version of the conclusion section of the manuscript. Thank you for the helpful suggestion.

Finally, I would like to invite the authors to include the abbreviation list of words at the end of this manuscript.

  • Regarding abbreviations, we followed the Journal instructions for the authors (https://www.mdpi.com/journal/pharmaceutics/instructions), which indicated that “Acronyms/Abbreviations/Initialisms” should be defined the first time when they appear in each of the sections”.

I recommend the acceptance of this manuscript after the authors performed the suggested corrections/additions. 

  • Thank you.

Reviewer 3 Report

This is very interesting study with different compounds (Baicalein, Baicalin, Metformin) to measure the antiradical/ antioxidative activities towards Antiglycation, Anti-α-Glucosidase and Antidiabetic Activities. Authors have performed several parameters to justify the study with possible mechanistic approaches. The presentation of the data in the manuscript is very impressive. I have some minor comments for the improvement of the manuscipt-

- In the abstract it is mention that "Successively, the ability of the compounds to pass through the cytoplasmic membranes of HT–29 cells was detected to estimate their potential absorption in vivo", if I am right this study is not obtained on animal model. This is quite confusing. Please correct.

- In the method section, it would be useful to add a flow diagram to indicated properly about the cells (in vitro) used in this study and the concentration used for the treatment along with incubation/ treatment time. 

- It would be nice to add a graphical abstract or a mechanistic figure summarizing the role of compounds used in this study as an inhibitor against radical formation. What is the possible mechanism? 

-In the result section, did authors see the overall effects of compounds considering at all concentrations in the statistics? Or just compared each concentration separately with the control group?

- In figure 2, a dot line maybe removed. This is not useful here. Maybe use a single line to indicate overall significant differences from all the 3 compounds with star.

-Method section should be improved. Atleast number of cells per well defined in each endpoints taken in this study. Or maybe add a separate paragraph in the method section on cell culture and treatment. For example, Line 505-506, mention the number of cells per wells.

Author Response

Response to Reviewer 3

Thank you very much for your appreciated review that helped us to improve the manuscript and also encouraged us in future researches. The suggestions have been taken into account, and all changes are reported with revision tracking in Word in the attached revised manuscript.

Comments and Suggestions for Authors

This is very interesting study with different compounds (Baicalein, Baicalin, Metformin) to measure the antiradical/ antioxidative activities towards Antiglycation, Anti-α-Glucosidase and Antidiabetic Activities. Authors have performed several parameters to justify the study with possible mechanistic approaches. The presentation of the data in the manuscript is very impressive. I have some minor comments for the improvement of the manuscript.

In the abstract it is mention that "Successively, the ability of the compounds to pass through the cytoplasmic membranes of HT–29 cells was detected to estimate their potential absorption in vivo", if I am right this study is not obtained on animal model. This is quite confusing. Please correct.

- Abstract L.21-22. We apologize for inconvenience. The sentence has been modified as suggested to clarify the meaning of the in vitro study with HT-29 cells, as reported in the revised manuscript.

In the method section, it would be useful to add a flow diagram to indicate properly about the cells (in vitro) used in this study and the concentration used for the treatment along with the incubation/treatment time.

-  Methods. Thank you for the suggestion. A flowchart was added, as Figure 11.

 It would be nice to add a graphical abstract or a mechanistic figure summarizing the role of compounds used in this study as an inhibitor against radical formation. What is the possible mechanism?

- A graphical abstract has been provided that helps to summarize the role of the compounds, relating to pharmacodynamic and pharmacokinetic studies.

In the result section, did authors see the overall effects of compounds considering at all concentrations in the statistics? Or just compared each concentration separately with the control group?

-  Results. As reported in the Statistical analysis section, one-way ANOVA, followed by the post-hoc Dunnett’s multiple comparison test, was used. The latter considers each experimental group’s mean against the control group mean [1].  Furthermore, in Figure 2 comparisons between the different treatments were made using Turkey’s test, which compared each experimental group with the other.

In figure 2, a dot line maybe removed. This is not useful here. Maybe use a single line to indicate overall significant differences from all the 3 compounds with star.

- The dotted line, which was added to the figures to indicate the maximal value of glycation without inhibitor, perhaps is not helpful as the reviewer suggested, thus, all dotted lines were removed from the figures.

Method section should be improved. At least number of cells per well defined in each endpoint taken in this study. Or maybe add a separate paragraph in the method section on cell culture and treatment. For example, Line 505-506, mention the number of cells per wells.

-The number of cells per well of uptake experiments (500,000 cells/well) was added in methods and a flowchart of the protocol has been added as reported in response to question no. 2 (Figure 11).

References

  1. McHugh, M. Lessons issue : in biostatistics Responsible writing in science Multiple comparison analysis testing in ANOVA. Biostats for Animal Science 2011, 21, 203–209.

Thank you.

Round 2

Reviewer 1 Report

Response to Reviewer 1

The authors thank the reviewer for the valuable suggestions in order to improve the scientific merit of the manuscript. Here, we provide detailed answers to all the raised questions.

“In this manuscript, the author investigated the therapeutic potentials of two natural compounds, baicalein and baicalin, for the treatment of diabetes mellitus (DM). By conducting a series of in vitro assays, the authors found that baicalein enables the reduction of glycation, α-glucosidase activity, and free radicals. On the other hand, baicalein is capable of reducing glycation and free radicals but not α-glucosidase activity. Besides, by using cell-based assay, the author found that baicalein has the ability to pass the cell membrane and concluded that this compound has potential absorption in vivo. While the author shows positive results for these two natural compounds when compared to current standard DM treatment drugs, including metformin and acarbose, some major concerns need to be addressed before considering them appropriate for publication.

First, the sections of this manuscript may need to be modified. Based on the instruction from the Journal, I believe it should be Introduction, Method, Result, and Discussion. Especially, the Result and Discussion should be in different sections. Moreover, it is expected that a manuscript will have a more detailed Discussion. Based on each finding in this manuscript, appropriate and more references should be provided for compare-and-contrast. Secondly, regarding the suitability of this manuscript to fit the submitted section “Pharmacokinetics and Pharmacodynamics”, it would be somehow difficult to relate the research conducted in this manuscript to the submitted section. Especially, one would expect the compounds investigated to be tested in an in vivo system. For example, there may have been published research that investigated glycation in mice, rats, or other species. Although the author did perform a cellular uptake assay and relate it to the bioavailability of baicalein, it may not be enough to indicate the pharmacokinetics and pharmacodynamics properties of the compound. Should the author provide some in vivo results, the manuscript may fit this section.

-  The authors thank the reviewer for comments. Regarding the sections of the manuscript, the Journal’s instructions for authors indicate that the Discussion section “may be combined with Results”, (https://www.mdpi.com/journal/pharmaceutics/instructions); therefore, in order to make easier for the reader to follow the entire data flow reported in the manuscript, we chose this option allowed for Research Manuscript type.

Response: Thank you for providing the link. Although the journal indicates that the Discussion section can be combined with the Result section, the manuscript may not meet the expectation of the Discussion as indicated in the Journal’s instructions. Although the author compared their results with others in some result sections, a more in-depth discussion and implication of each result are missing. A broad perspective and relevant references should be provided instead of mainly presenting the results of the experiment. In addition, the author did not mention the limitations in this manuscript.  

Regarding the issue on the suitability of the manuscript to fit the section “Pharmacokinetics and Pharmacodynamics”, we believe that the plan of the in vitro experiments, considering the enzyme kinetics and cell uptake studies, can fit both the Pharmacokinetics and the Pharmacodynamics topics. It was not our aim to investigate in vivo properties of the compounds that could represent a future advancement of the research, based on the present results obtained in specific in vitro conditions. Furthermore, baicalein and baicalin are already used as components of medicinal plant extracts or food supplements, and thus the detection in simple pharmacokinetics models of cellular permeability could be useful to support the current use of these products.

Response: Thank you for the explanation. I acknowledge that the author states that the enzyme kinetics and cell uptake studies would fit the PK and PD section. To facilitate interpreting the results by the readers, the author may at least explain how these results can affect drug PK (i.e., ADME) and PD (i.e., efficacy), on which the current manuscript has little information.

Third, it would not be easy to comprehensively evaluate the antiglycation capability of the investigated compounds based on the individual in vitro assays. It would be preferable if the author could provide figures/schemes outlining the pathway of AGE formation and how each in vitro assay correlated to the corresponding steps. Then, the author may consider mentioning steps or properties within the AGE pathway that were not tested and their contributions to the final evaluation of these two compounds.

-  A scheme describing the pathway of AGE formation and the corresponding steps evaluated with each in vitro assay has been introduced as a supplementary figure (Figure S1).

Response: Thank you for providing the figure.

Fourth, would the author indicate the study objective more clearly? The objective of the study should be mentioned in the last part of the Introduction and in the abstract. Since the study objective is not clearly defined, it is quite confusing why the author compared the investigated compounds with metformin and acarbose. In addition, would the author mention the innovation of the current study? For each result in this manuscript, it seems the author mainly provided references and commented on whether their finding corresponded to the previous reports. The knowledge added to the field should be highlighted.

-  A better explanation of the critical suggested points has been provided in the manuscript. As suggested, the purpose of this research has been declared in the last part of the Introduction and in the abstract. However, for the abstract the typographical rules of the Journal require a total of about 200 words maximum; thus, the abstract was only slightly changed.

Response: Thank you for the minor change in the abstract and in the Introduction.

Minor comments:

The data should be presented as mean and standard deviation (SD) instead of SEM.

-  We thank the reviewer for comment; however, we used standard error of the mean (SEM) as an inferential parameter of the sampling process in our experimental design. The aim was to give information about the precision of the data, and how well the sample can represent the entire population of data, since SEM provides a description of how far is the sample mean from the population mean; the use of SEM and respective inferential error bars (see Cumming et al., Journal of Cell Biology, 177 (1), 7-11, 2007) helps in comparing samples from two groups, to see if they are different. On the other hand, there is less interest here about the degree to which single measurements within the sample differ from the sample mean, as indicated by standard deviation (SD).

Response: thank you for the reference. It would be good for the author to provide n of each experiment, and thus the variability of the data can be calculated.

Figure 10 is missing.

- We are sorry for the mistake, due to a preliminary figure numbering. We fixed the problem in the revised version. Furthermore, the figures were updated graphically, improving their legibility.

Figures 2 and 3: Please explain why the fluorescence values of BSA+glucose/glyoxal/ribose are different between Figures 2A and 3 (dashed line).

-  The fluorescence values of BSA+glucose/glyoxal/ribose are different between Figures 2A and 3 because they are from different experiments; in a first phase, the maximal glycation was studied alone using different days of incubation. Successively, in a second phase, glycation experiments were performed using the inhibitors. In each set of experiments, we used their own controls. For this reason, the absolute fluorescence data are relatively dissimilar.

Response: thank you for the explanation. Please add the description for clarification in the manuscript.

Figures 3A and 3B provided similar information and may seem duplicated.

-  Figure 3A (now 4A) shows the absolute values of GK cross-linking, together with the reference values (GK peptide + ribose and AG positive control), while Figure 3B (now 4B) reports the normalized percentage data. We believe that both panels can provide useful information.

Figure 6: Please describe the definition of “v versus”

-        (Now is Figure 7) We provided a detailed explanation of the term, that is, the enzyme activity (v) versus α–glucosidase enzyme concentration.

Table 1: the footnotes of a and b are missing.

- The superscript letters refer to the “superscript method” used in statistical analysis, which is a current way of expressing multiple pairwise comparisons (see, for example, discussion by H.P. Piepho doi:10.2134/agronj2017.10.0580). The letter display allows an efficient reporting method of statistical significance, that permits one to avoid the use of an excessive number of different symbols, as occurring with post hoc statistical tests of significance, that perform all the possible paired combinations, also considering the family-wise error rate. Therefore, the interpretation of these superscripts in Table 1 is, as reported in the footnotes: “Values not sharing a common superscript letter differ significantly at p< 0.0001.”

Response: thank you for the reply.

Author Response

Response to Reviewer 1 

The authors thank the reviewer for the valuable suggestions in order to improve the scientific merit of the manuscript. Here, we provide detailed answers to all the raised questions. 

“In this manuscript, the author investigated the therapeutic potentials of two natural compounds, baicalein and baicalin, for the treatment of diabetes mellitus (DM). By conducting a series of in vitro assays, the authors found that baicalein enables the reduction of glycation, α-glucosidase activity, and free radicals. On the other hand, baicalein is capable of reducing glycation and free radicals but not α-glucosidase activity. Besides, by using cell-based assay, the author found that baicalein has the ability to pass the cell membrane and concluded that this compound has potential absorption in vivo. While the author shows positive results for these two natural compounds when compared to current standard DM treatment drugs, including metformin and acarbose, some major concerns need to be addressed before considering them appropriate for publication.

 First, the sections of this manuscript may need to be modified. Based on the instruction from the Journal, I believe it should be Introduction, Method, Result, and Discussion. Especially, the Result and Discussion should be in different sections. Moreover, it is expected that a manuscript will have a more detailed Discussion. Based on each finding in this manuscript, appropriate and more references should be provided for compare-and-contrast. Secondly, regarding the suitability of this manuscript to fit the submitted section “Pharmacokinetics and Pharmacodynamics”, it would be somehow difficult to relate the research conducted in this manuscript to the submitted section. Especially, one would expect the compounds investigated to be tested in an in vivo system. For example, there may have been published research that investigated glycation in mice, rats, or other species. Although the author did perform a cellular uptake assay and relate it to the bioavailability of baicalein, it may not be enough to indicate the pharmacokinetics and pharmacodynamics properties of the compound. Should the author provide some in vivo results, the manuscript may fit this section.

  Authors’ reply (2) The authors thank the reviewer for comments. Regarding the sections of the manuscript, the Journal’s instructions for authors indicate that the Discussion section “may be combined with Results”, (https://www.mdpi.com/journal/pharmaceutics/instructions); therefore, in order to make easier for the reader to follow the entire data flow reported in the manuscript, we chose this option allowed for Research Manuscript type.

Reviewer: Thank you for providing the link. Although the journal indicates that the Discussion section can be combined with the Result section, the manuscript may not meet the expectation of the Discussion as indicated in the Journal’s instructions. Although the author compared their results with others in some result sections, a more in-depth discussion and implication of each result are missing. A broad perspective and relevant references should be provided instead of mainly presenting the results of the experiment. In addition, the author did not mention the limitations in this manuscript.  

Authors’ reply (2) We chose to combine Results with Discussion, since the quite abundant amount of the experimental data would have required a too complex and extensive continuous reference in a separate Discussion section. Instead, by on commenting progressively each group of results, the interpretation is made more directly to the reader. Regarding limitations, this is an in vitro study representing a picture of the overall properties of the investigated compounds, which could also be further studied, successfully, by other authors, on both animal and/or human subjects.

Regarding the question of the suitability of the manuscript to fit the section ‘Pharmacokinetics and Pharmacodynamics”, we believe that the present in vitro experiments, consisting of glycation study, enzyme kinetics and cell uptake studies, can fit both the Pharmacokinetics and the Pharmacodynamics topics. It was not our aim to investigate the in vivo properties of the compounds that could represent a future advancement of the research based on the present results obtained under specific in vitro conditions. Furthermore, baicalein and baicalin are already used as components of medicinal plant extracts or food supplements, and thus the detection in simple pharmacokinetics models of cellular permeability could be useful to support the current use of these products. 

Response: Thank you for the explanation. I acknowledge that the author states that the enzyme kinetics and cell uptake studies would fit the PK and PD section. To facilitate interpreting the results by the readers, the author may at least explain how these results can affect drug PK (i.e., ADME) and PD (i.e., efficacy), on which the current manuscript has little information.

Authors’ reply (2) We thank the Reviewer for additional comments regarding this aspect of the manuscript. We would like to highlight the fact that the special issue of the Journal entitled “Recent advances in natural Products”, according to the information provided to the contributors, welcomes research which may add knowledge about the many aspects linked to natural compounds from plant and animal sources, including, although not limited, the aspects of PK and PD. As we have already explained, we believe that our in vitro research, considering glycation assessment, enzyme kinetics and cell uptake studies, can cover the topic, as suggested by the invitation of the Journal. 

Third, it would not be easy to comprehensively evaluate the antiglycation capability of the investigated compounds based on the individual in vitro assays. It would be preferable if the author could provide figures/schemes outlining the pathway of AGE formation and how each in vitro assay correlated to the corresponding steps. Then, the author may consider mentioning steps or properties within the AGE pathway that were not tested and their contributions to the final evaluation of these two compounds.

 -  A scheme describing the pathway of AGE formation and the corresponding steps evaluated with each in vitro assay has been introduced as a supplementary figure (Figure S1).

Response: Thank you for providing the figure.

Fourth, would the author indicate the study objective more clearly? The objective of the study should be mentioned in the last part of the Introduction and in the abstract. Since the study objective is not clearly defined, it is quite confusing why the author compared the investigated compounds with metformin and acarbose. In addition, would the author mention the innovation of the current study? For each result in this manuscript, it seems the author mainly provided references and commented on whether their finding corresponded to the previous reports. The knowledge added to the field should be highlighted.

 -  A better explanation of the critical suggested points has been provided in the manuscript. As suggested, the purpose of this research has been declared in the last part of the Introduction and in the abstract. However, for the abstract the typographical rules of the Journal require a total of about 200 words maximum; thus, the abstract was only slightly changed.

Response: Thank you for the minor change in the abstract and in the Introduction.

 Minor comments:

The data should be presented as mean and standard deviation (SD) instead of SEM.

-  We thank the reviewer for comment; however, we used standard error of the mean (SEM) as an inferential parameter of the sampling process in our experimental design. The aim was to give information about the precision of the data, and how well the sample can represent the entire population of data, since SEM provides a description of how far is the sample mean from the population mean; the use of SEM and respective inferential error bars (see Cumming et al., Journal of Cell Biology, 177 (1), 7-11, 2007) helps in comparing samples from two groups, to see if they are different. On the other hand, there is less interest here about the degree to which single measurements within the sample differ from the sample mean, as indicated by standard deviation (SD).

Response: thank you for the reference. It would be good for the author to provide n of each experiment, and thus the variability of the data can be calculated.

 Authors’ reply (2): The manuscript already states, in the legend of each figure, the number of replicates for each experiment. Anyway, a measure of the data variability can be deduced from the reported SEM and respective inferential error bars.

Figure 10 is missing.

- We are sorry for the mistake, due to a preliminary figure numbering. We fixed the problem in the revised version. Furthermore, the figures were updated graphically, improving their legibility.

 Figures 2 and 3: Please explain why the fluorescence values of BSA+glucose/glyoxal/ribose are different between Figures 2A and 3 (dashed line).

 -  The fluorescence values of BSA+glucose/glyoxal/ribose are different between Figures 2A and 3 because they are from different experiments; in a first phase, the maximal glycation was studied alone using different days of incubation. Successively, in a second phase, glycation experiments were performed using the inhibitors. In each set of experiments, we used their own controls. For this reason, the absolute fluorescence data are relatively dissimilar.

 Response: thank you for the explanation. Please add the description for clarification in the manuscript.

Authors’ reply (2) Thank you for the suggestion. Each figure refers to its own control being from different experimental sets; this is true for each type of protocols. Thus, we think that the information is not useful for reading the results and, therefore, we did not add additional explanation.

 Figures 3A and 3B provided similar information and may seem duplicated.

 -  Figure 3A (now 4A) shows the absolute values of GK cross-linking, together with the reference values (GK peptide + ribose and AG positive control), while Figure 3B (now 4B) reports the normalized percentage data. We believe that both panels can provide useful information.

 Figure 6: Please describe the definition of “v versus”

 -  (Now is Figure 7) We provided a detailed explanation of the term, that is, the enzyme activity (vversus α–glucosidase enzyme concentration.

Table 1: the footnotes of a and b are missing.

 - The superscript letters refer to the “superscript method” used in statistical analysis, which is a current way of expressing multiple pairwise comparisons (see, for example, discussion by H.P. Piepho doi:10.2134/agronj2017.10.0580). The letter display allows an efficient reporting method of statistical significance, that permits one to avoid the use of an excessive number of different symbols, as occurring with post hoc statistical tests of significance, that perform all the possible paired combinations, also considering the family-wise error rate. Therefore, the interpretation of these superscripts in Table 1 is, as reported in the footnotes: “Values not sharing a common superscript letter differ significantly at p< 0.0001.”

Response: thank you for the reply.

Authors’ reply (2) Thank you for the review and suggestions.
